# Seismic Performance and Optimization of a Novel Partial Seismic Isolation System for Frame Structures

**Baokui Chen, Yuxin Qiu, Jingang Xiong \*, Yaru Liu and Yanqing Xu**

School of infrastructure Engineering, Nanchang University, Nanchang 330031, China; bkchen@ncu.edu.cn (B.C.); 416000210055@email.ncu.edu.cn (Y.Q.); 411114619033@ncu.edu.cn (Y.L.); xuyanqing@ncu.edu.cn (Y.X.)
\* Correspondence: xiongjingang@ncu.edu.cn

**Abstract:** To improve the safety performance of important rooms, such as operating rooms and disaster command centers, during an earthquake, a novel partial seismic isolation system suitable for new and existing frame structures is proposed, and the seismic and optimization analysis is carried out. Using the finite element numerical simulation method, the models of the ordinary frame structure and the partial isolated system structures were established. Considering the seismic response of the isolation room, the design safety of the partial isolation room, and the seismic impact on the overall structure, this study analyzed the damping effect of the partial isolation system. We changed the type of isolation bearing, the location of the isolation room, and the load to further optimize the calculation of the seismic isolation structure. The results show that the new partial isolation system could significantly reduce the seismic response of the isolated room under the action of a magnitude-8 rare earthquake. The damping rate of the relative acceleration and relative displacement between the top and bottom of the columns of the isolated room could reach 90%. It was found that the partial seismic isolation system proposed in this paper was applicable to reinforced concrete frame structures and could significantly reduce the seismic response of the isolated rooms without affecting the seismic performance of the main building. This partial seismic isolation system is easy to construct, applicable to both existing and new structures, and provides a new and effective seismic mitigation measure to improve the seismic performance of locally important rooms in the structure.

**Keywords:** partial seismic isolation; frame structure; numerical simulation; seismic performance; isolation bearing

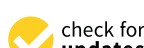



## 1. Introduction

Ensuring the safety of all equipment and personnel in some rooms with important functions, such as the operating room, and not affecting the normal use function of the room an under earthquake, is one of the important fields of seismic research for building structures. Relevant specialists have presented some novel ideas and initiatives. Tan et al. [1] presented a partial seismic isolation system for master−servant coupled structures, in which a seismic isolation layer is only installed in the servant structure and the servant structure is sacrificed during strong earthquakes to protect the master structure's safety. Morales et al. [2] presented a new low-cost seismic protection system that uses recycled automobile rubber tires to isolate certain rooms or equipment in buildings in order to minimize the dynamic response and to enhance structural and member performance. However, complex evaluation of the tire's characteristics is required before the seismic protection system can be implemented and deployed in critical health care facilities. Losanno et al. [3] investigated the use of recycled rubber by focusing on modeling aspects [4] and aging effects [5], and developed a sustainable foundation isolation system based on a new low-cost isolator. The results show that the device can effectively reduce the structure's absolute acceleration and base shear, indicating that this low-cost isolation device has the potential to reduce earthquake risk in developing countries. Baggio et al. [6] investigated a double

concave curved surface slider (DCCSS) seismic isolation device for sculptural seismic protection and evaluated the effectiveness of this isolation system using nonlinear dynamic analysis. On the contrary, Pellecchia et al. [7,8] proposed the use of elastomeric bearings to protect art-objects from earthquake-induced vibrations in order to contribute addressing similar issues in the challenging task of protecting cultural heritage from earthquake damage. Mezghani et al. [9] improved the metallic dampers to offer a higher performance to protect sensitive equipment under moderately strong or strong earthquakes, then proposed the wire mesh vibration damper (WMVD) for vibration-sensitive equipment. The results revealed that the WMVD isolated system can effectively attenuate a seismic response of more than 85%. Meanwhile, floor isolation systems have been becoming increasingly popular as a protective measure for nonstructural components. Jia et al. [10] optimized the floor isolation system based on the reliability criterion to maximize the probability that the acceleration response of the protected equipment would not exceed the acceptable performance limit. In addition, global sensitivity analysis based on samples was integrated to study the importance of different risk factors regarding system failure probability.

Theoretical and experimental research on seismically isolated buildings has received increased attention in recent years, and related research has become increasingly comprehensive [11–17]. To strengthen a medical building, Ye et al. [18] used three seismic isolation schemes: foundation isolation, additional flexural restraint bracing, and additional sway walls. They discovered that the seismic isolation scheme could reduce both the displacement and absolute acceleration responses of the structure, which has obvious benefits in reducing the economic losses from earthquakes. Murota et al. [19] used numerical and experimental methods to explore the suitability of high damping rubber bearings in the seismic isolation of residential buildings in Turkey, evaluating the seismic response of the buildings and determining the efficacy of the seismic isolation system. Sung et al. [20] proposed incorporating an elliptical member equipped with a rubber cylinder in a portal reinforced concrete frame and conducted shaking table tests, which showed that the proposed strengthening method could not only restore the seismic capacity, but also improve the seismic resistance of the reinforced concrete frame damaged by the earthquake. Zheng et al. [21] created a scaled-down model of a four-story frame structure with friction pendulum support for seismic isolation and conducted shaking table tests, which revealed that the friction pendulum can significantly reduce inter-floor displacement and floor acceleration, as well as provide good seismic isolation. Xu et al. [22] suggested an SMA-based self-resetting bracket, and the finite element analysis revealed that the bracket with a super elastic SMA bolted connection has a strong self-resetting capability, and can significantly reduce the residual deformation of the structure after the seismic response. Yang et al. [23] constructed a theoretical mechanical model of an oblique rotating three-dimensional seismic isolation device, and conducted static tests and numerical simulation studies on it, concluding that the bearing can guarantee the bearing capacity with vertical displacement and can achieve the vertical energy dissipation seismic isolation goal. Isolation systems mainly rely on energy dissipation mechanisms, usually using the concept of viscous damping to evaluate these energy dissipation mechanisms. Li et al. [24] studied the equivalent problem of friction and viscous damping of a spring friction pendulum vibration isolation system under sine wave ground motion, providing a new method for unifying the concept of damping and evaluating the amount of damping in structures.

Although both base and floor seismic isolation techniques are established for new structures [25–30], overall seismic isolation is costly [31] and difficult to apply to existing structures. For operating rooms and other local functional rooms with special damping requirements, it is of great significance to reduce the seismic response of local rooms and to ensure equipment and personnel work sustainably in the room after an earthquake or even at the epicenter. At present, there is research on partial isolation considering a floor isolation system [32], but it mainly focuses on the isolation of equipment. This can only ensure the equipment is intact, but it cannot protect other ancillary components, and cannot guarantee the continuity and safety of important work such as surgery in earthquakes. Therefore,

a novel partial seismic isolation structural system based on foundation seismic isolation is proposed, which could not only play an isolation and damping role for important equipment, but could also ensure the safety of equipment and non-structural components of the whole room, so that the functional room can maintain its complete functionality during and after an earthquake. This isolation system can be used for partial seismic isolation retrofit of rooms that require key fortification, such as operating rooms and intensive care units. In order to determine the effectiveness and safety of the proposed local seismic isolation system, through finite element numerical simulation technology, this paper systematically analyzes the seismic reduction effect and safety of the new isolation system, as well as the impact on the overall structural seismic performance. The new system proposed in the study can not only reduce the seismic response of the local structure, but through optimization analysis, it can also gradually form a local seismic isolation design method applicable to different functional objectives. The research results will expand the ideas of the research on the seismic performance of local structures and will propose an innovative design method for local structure seismic reduction, which has important research significance and engineering value.

## 2. Partial Seismic Isolation System and Numerical Model

### 2.1. Partial Seismic Isolation System

This study proposes a partial seismic isolation system based on the foundation isolation for both existing and new frame structures, as shown in Figure 1. The partial seismic isolation system is composed of structural columns, upper and lower ring beams, floor slabs, and maintenance members, which is connected to the main beam of the frame structure through an isolation bearing. Furthermore, the isolation joints between two sides and adjacent columns are 200 mm. The seismic isolation bearing can be set at the position of the frame structure's main beam, and the partial seismic isolation system does not come into contact with the structural frame columns or the original structural floor slab, retaining a suitable safety distance.

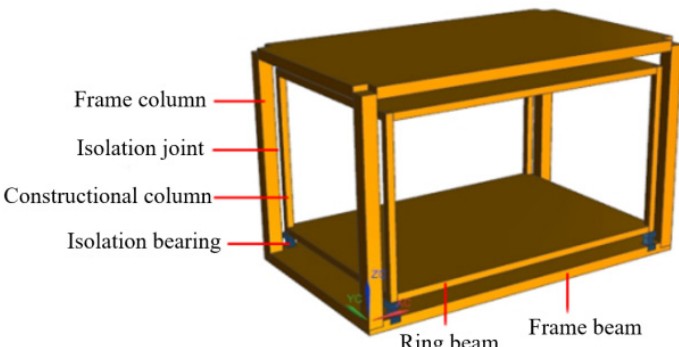

**Figure 1.** Novel partial seismic isolation structure diagram.

The partial seismic isolation system achieves a flexible connection in the horizontal direction between the original frame beam and the partial room through the seismic isolation bearing. The seismic energy transferred to the partial isolation room is significantly reduced and changes the dynamic characteristics of the room to approximate whole translation motion, thereby reducing the seismic response of the partial room [33]. This is because under the action of an earthquake, the bearing consumes energy because of the hysteretic behavior, so that the energy transmitted to the isolated room is significantly reduced, thus leaving it in a whole translation motion. As shown in Figure 2, this study analyzes the performance and effect of the partial seismic isolation system by comparing the seismic response of the original structure and the partial isolation structure.

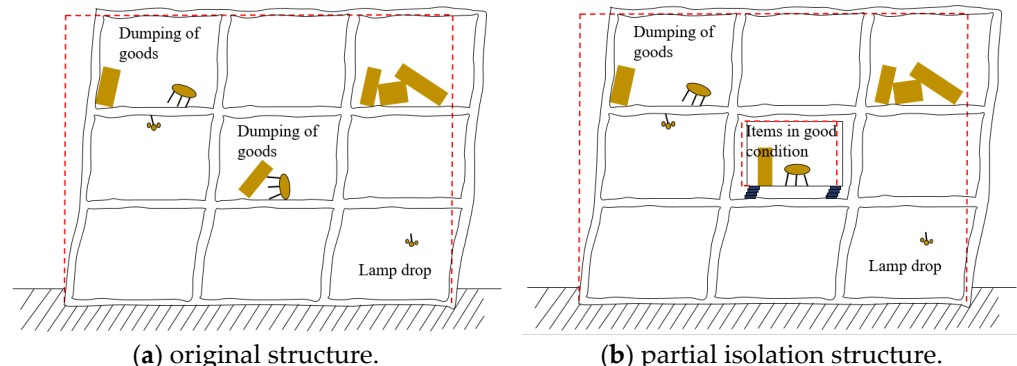

(**a**) original structure.　　(**b**) partial isolation structure.

**Figure 2.** Response of structures under earthquake.

*2.2. Numerical Model*

In order to reduce the interference of building construction and other factors on the seismic response of the partial seismic isolation system, the study takes a four-story frame structure with relatively simple construction as the research object and discusses the seismic reduction effect of the partial seismic isolation system.

The numerical model was established using finite element analysis software SAP2000, and the original concrete frame structure model is shown in Figure 3. At the same time, a frame model using the novel partial seismic isolation system was established, in which the seismic isolation room was set in the middle of the second floor, and the rest of the structure was the same as the original structure. The basic layout of the building was as follows: there were five spans in the X-direction with a span of 6 m and three spans in the Y-direction with a span of 4 m. The structure had four floors and the height of each floor was 3.6 m. The cross-sectional dimensions of the columns were 700 mm × 700 mm, the main beam was arranged in an X-direction with a cross-sectional dimension of 300 mm × 700 mm, and the secondary beam was arranged in a Y-direction with a cross-sectional dimension of 300 mm × 600 mm. The concrete compressive strength of the beams and columns was 30 MPa, and the yield strength of longitudinal ribs and stirrup were 335 MPa and 300 MPa respectively. The constant load of the floor was 3 kN/m$^2$ and the live load was 2 kN/m$^2$. The seismic precautionary intensity of the structure was magnitude 8, and the corresponding design basic acceleration of the ground motion was 0.2 g. The beam−column units in the frame structure were input according to the corresponding frame section and material properties, and the beam−column nodes were set as rigid junctions; the floor slab was input using shell units and by specifying the partition bindings to achieve the assumption of the infinite in-plane stiffness of the floor slab [34].

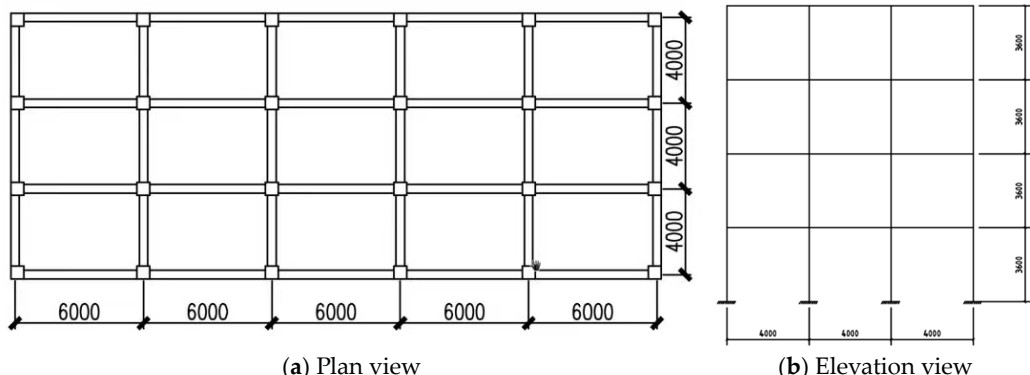

(**a**) Plan view　　　　　　　　　　　　(**b**) Elevation view

**Figure 3.** *Cont.*

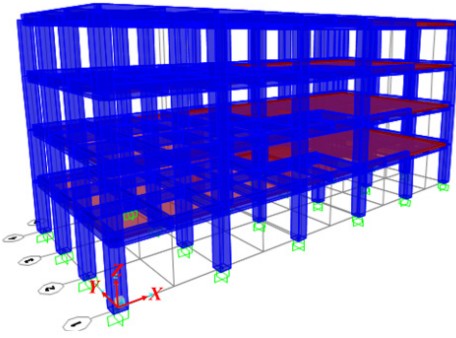

(**c**) Finite element model

**Figure 3.** Layout drawing and finite element model of the original frame structure.

### 2.3. Selection of Seismic Isolation Bearing

In this project, the partial isolation room was set on the main beam only through the seismic isolation bearing, and the specific size of the bearing was determined by the reaction force at the bottom of each column and the bearing surface pressure under gravity load, assuming that the main beam of the original structure had a sufficient support capacity. The relevant mechanical properties of the selected seismic isolation bearing are listed in Table 1.

**Table 1.** Basic parameters of the lead rubber bearing.

| Bearing Type | Effective Diameter (mm) | Total Rubber Thickness (mm) | Pre-Yield Stiffness (kN/m) | Equivalent Stiffness (kN/m) | Vertical Stiffness (kN/mm) | Yield Force (kN) |
|---|---|---|---|---|---|---|
| LRB300 | 300 | 56 | 6440 | 760 | 1100 | 16 |

The nonlinear hysteresis curve of the LRB isolation lead core rubber support could be simplified to the bilinear model shown in Figure 4 [35]. In Figure 4, the pre-yield stiffness $K_{b1}$, post-yield stiffness $K_{b2}$, and equivalent horizontal shear stiffness $K_{eq}$ of the seismic isolation bearing were calculated as follows.

$$K_{b1} = \frac{Q_y}{X_y} \tag{1}$$

$$K_{b2} = \frac{(Q_b - Q_y)}{(X_b - X_y)} \tag{2}$$

$$K_{eq} = \frac{Q_b - Q_a}{X_b - X_a} \tag{3}$$

where $X_b$ is the maximum horizontal positive displacement, $X_a$ is the maximum horizontal negative displacement, $Q_b$ is the horizontal shear force corresponding to $X_b$, and $Q_a$ is the horizontal shear force corresponding to $X_a$.

### 2.4. Seismic Waves

Considering the site category, fortification intensity, and seismic wave selection principle of the model, the EL-Centro wave, Taft wave, and one artificial wave were selected. Selecting the appropriate peak ground acceleration (PGA) of seismic waves is a key step in structural seismic response analysis [36,37]. The seismic waves were taken as the horizontal bidirectional input, and the X-direction peak acceleration was adjusted to 400 gal (the magnitude 8 rare earthquake level in Chinses code [38]). The peak acceleration curves of the input were adjusted according to the ratio of X:Y = 1:0.85 [38], respectively. The adjusted X-direction acceleration time history of the seismic wave is shown in Figure 5. The comparison of the seismic response spectrum and standard response spectrum is shown in Figure 6. The empirical response spectrum of the seismic record is consistent with the

statistical significance of the normative response spectrum, so the selected seismic wave time curve met the selection requirements.

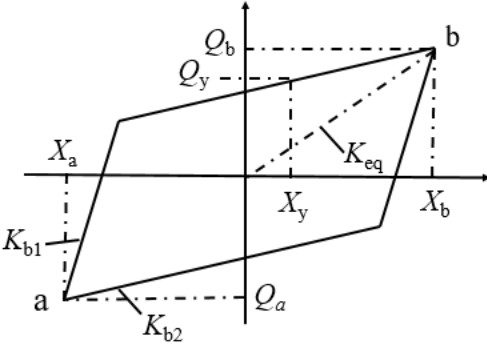

**Figure 4.** Bilinear hysteretic model of the isolated bearing.

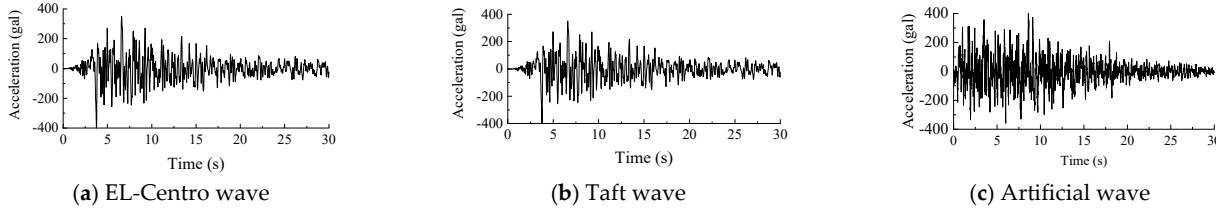

**Figure 5.** X−direction time−history curve of the seismic waves.

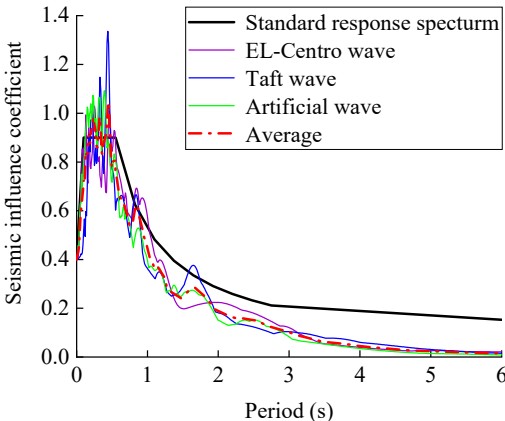

**Figure 6.** Comparison of the seismic response spectrum and standard response spectrum.

## 3. Analysis of Seismic Isolation Effect

### 3.1. Modal Analysis

The modal analysis can qualitatively determine the seismic response of the structure, which is the basis for other dynamic analyses. The natural periods of the original structure and the partial isolation structure are detailed in Table 2. Table 3 shows the natural periods of the isolated room in the partial isolated system.

**Table 2.** Modal analysis data of the structure.

| Mode of Vibration | Original Structure | | | | Isolated Structure | | | |
|---|---|---|---|---|---|---|---|---|
| | Period (s) | UX | UY | RZ | Period (s) | UX | UY | RZ |
| 1 | 0.413 | 0.820 | 0 | 0 | 0.413 | 0.798 | 0 | 0 |
| 2 | 0.383 | 0 | 0.826 | 0 | 0.384 | 0 | 0.804 | 0 |
| 3 | 0.340 | 0 | 0 | 0.830 | 0.340 | 0 | 0 | 0.825 |
| 4 | 0.124 | 0.118 | 0 | 0 | 0.133 | 0 | 0 | 0 |
| 5 | 0.117 | 0 | 0.116 | 0 | 0.132 | 0 | 0 | 0 |

**Table 3.** Modal analysis data of the partial isolation room.

| Mode of Vibration | Period (s) | UX | UY | RZ |
|:---:|:---:|:---:|:---:|:---:|
| 1 | 0.669 | 0 | 0.025 | 0 |
| 2 | 0.668 | 0.025 | 0 | 0 |
| 3 | 0.486 | 0 | 0 | 0.001 |

From Table 2, it can be seen that the vibration periods of the isolated structure and the original structure were almost the same, indicating that the partial isolation system did not affect the self-oscillation characteristics of the overall structure. UX, UY, and UZ were the mass participation ratios in the X, Y, and Z directions, respectively, of the vibration type, and the analysis shows that the first order of the structure was whole translation motion in the X direction (UX + UY > UZ), the second order was whole translation motion in the Y direction, and the third order was a torsional vibration type around the Z axis (UX + UY < UZ). The first three orders of the period of the partial isolation room were 0.669 s, 0.668 s, and 0.486 s, which were significantly larger than the first three orders of the self-oscillation period of the original structure, namely, 0.413 s, 0.380 s, and 0.340 s. The self-oscillation period of the partial isolation room was significantly longer, which was conducive to avoiding the high frequency zone of seismic waves and greatly reducing the seismic energy transferred to the isolated room, thus improving the safety of the partial isolation room.

### 3.2. Dynamic Properties of Seismically Isolated Rooms

The three selected seismic waves were input to each model and the dynamic history analysis under a magnitude 8 rare earthquake was carried out. It was found that the X-direction seismic response was larger compared with the Y-direction, so the envelope value of the X-direction response of the structure under the action of the three seismic waves was taken as a representative value for the nonlinear time−history analysis. A reasonable evaluation index, the damping rate ($\Delta$), was chosen to assess the damping effect of each seismic isolation structural model using the following equation:

$$\Delta = \frac{(REP_{ori} - REP_{iso})}{REP_{ori}} \times 100\% \tag{4}$$

where $REP_{ori}$ is the seismic response of the original structure and $REP_{iso}$ is the seismic response of the isolated structure.

Figure 7 gives the relative acceleration and displacement time histories of the partial isolation room for the two structural systems under magnitude 8 rare earthquakes, i.e., the relative acceleration and relative displacement between the top and bottom of the columns of the isolated room. As shown in the figure, the seismic response of the room with a new partial isolation structural system is significantly smaller than that of the original structure. The maximum values of the relative acceleration before and after seismic isolation were 4.98 m/s$^2$ and 0.46 m/s$^2$, respectively, and the maximum values of relative displacement were 19.93 mm and 2.07 mm, respectively, which were calculated from Equation (4). The seismic isolation system changed the dynamic characteristics of the isolated room through the seismic isolation bearing. The isolated room showed the whole translation motion, and its seismic response was effectively reduced.

### 3.3. Hysteresis Performance of the Vibration Isolation Bearing

The choice of seismic isolation bearing has an important influence on the seismic isolation effect. Figure 8 shows the hysteresis curve of the seismic isolation bearing. Under the effect of a magnitude 8 rare earthquake, the maximum horizontal displacement of the bearing was 53 mm. The bearing compressive stress was 1.27 MPa, which was much less than the specification requirement. At the same time, the hysteresis ring of the bearing

was full, indicating that the partial seismic isolation system had a good hysteresis energy dissipation performance.

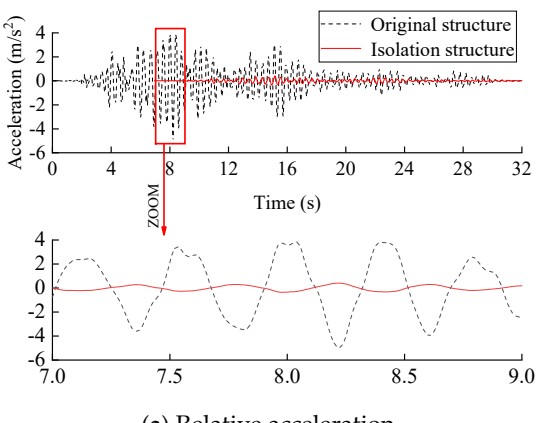

(**a**) Relative acceleration

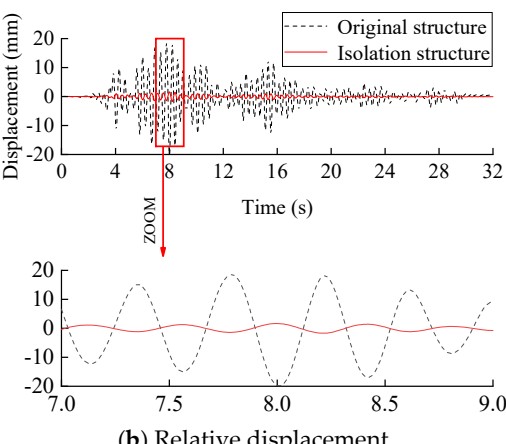

(**b**) Relative displacement

**Figure 7.** Damping effect of an isolated room.

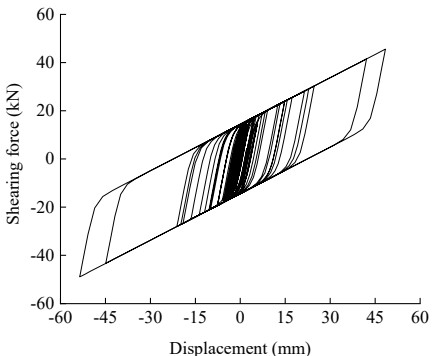

**Figure 8.** Hysteretic curve of the isolated bearing.

### *3.4. Safety of Partial Seismic Isolation Systems*

During the seismic response of the partial isolation rooms, collisions with the original structural frame columns should be avoided. Therefore, the study analysed the variation of the distance between the structural columns of the partial isolation room and the adjacent frame columns under seismic loading, as a means of determining the safety of the structure. Figure 9 shows the displacement time curve between the frame column of the seismically isolated room and the adjacent members with an initial distance of 200 mm, which was the width of the reserved seismic isolation joint. Non-positive spacing means a collision occurred. The minimum distance between the partial isolation room and the adjacent members was 167 mm under a rare earthquake of magnitude 8, indicating that no collision would occur between the partial isolation room and the adjacent members, and that the partial isolation structure had a high safety reserve. At the same time, the shear force and bending moment of the beam below the bearing were calculated to be less than its cross-sectional load carrying capacity.

### *3.5. Effect of Partial Seismic Isolation on the Structure as a Whole*

The seismic response of the original structure and the top of the partial isolation structure, as well as the inter-story displacement angles and shear forces, are shown in Figure 10 to Figure 11, respectively. Because of the small effect of the partial isolation system on the change in stiffness of the overall structure and the limited mass of the isolated rooms, by comparing the overall structural response of the two models, it was found that the use of partial seismic isolation or not had little effect on the overall seismic performance of the structure. The maximum inter-story displacement angle of the partial isolation structure

was 1/180 under a rare earthquake of magnitude 8, which still met the code requirement of 1/50 for the elastic−plastic inter-story displacement angle under a rare earthquake. It was thus judged that the partial seismic isolation system had no effect on the seismic response and stability of the overall frame structure.

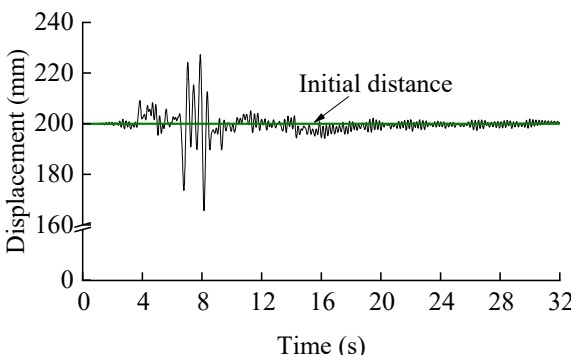

**Figure 9.** Distance between the isolated room column and adjacent column.

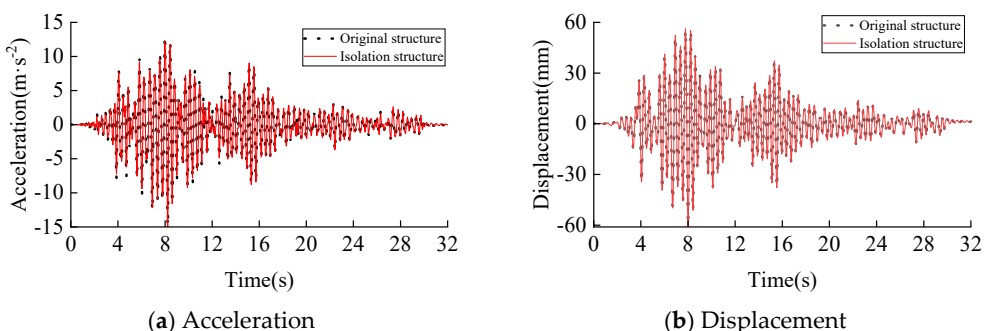

**Figure 10.** Acceleration and displacement time−history curve of the structure top.

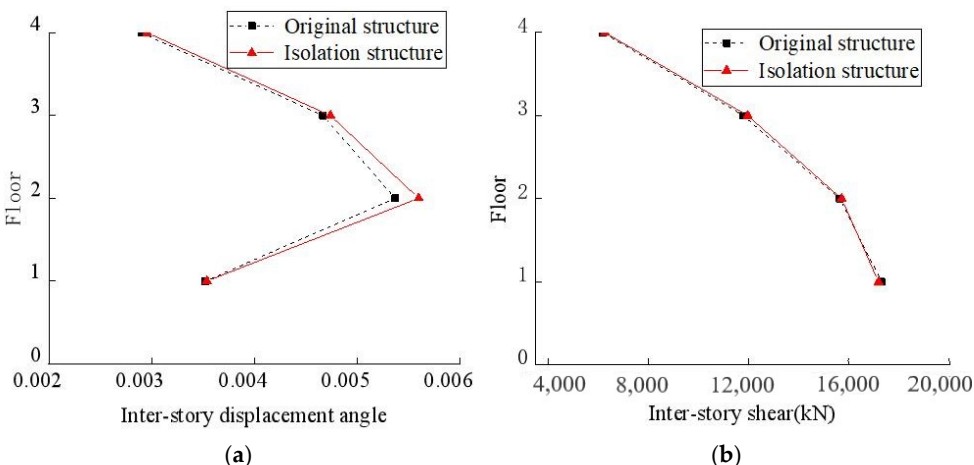

**Figure 11.** (**a**) Inter−story displacement angle and (**b**) inter-story shear force.

## 4. Optimal Design of Partial Seismic Isolation Systems

### 4.1. Influence of Bearing Type on Vibration Damping in Isolated Rooms

To further optimize the partial seismic isolation system and improve the seismic performance of the structure, the effect of different bearing types on the seismic performance of the structure was investigated. The study added a partial seismic isolation model using natural rubber bearings, which was the same as the partial seismic isolation structural model in Section 2 except for the bearings. The relevant mechanical performance parameters of the natural rubber bearings used are shown in Table 4.

**Table 4.** Basic parameters of the linear natural rubber bearing.

| Type of Bearing | Effective Diameter (mm) | Total Rubber Thickness (mm) | Equivalent Stiffness (kN/m) | Vertical Stiffness (kN/mm) |
|---|---|---|---|---|
| LNR300 | 300 | 56 | 490 | 900 |

By comparing the analysis results of the two bearing models with the original frame structure, the seismic response of the partial isolation room is shown in Figure 12. The relative peak accelerations of the partial isolation room under the lead-core rubber bearing and natural rubber isolation bearing were 0.46 m/s$^2$ and 0.51 m/s$^2$, and the relative displacement peaks were 2.07 mm and 2.52 mm, respectively. Both of them were far less than the partial room seismic response of the original structure, and the lead rubber bearing was better than the natural rubber bearing.

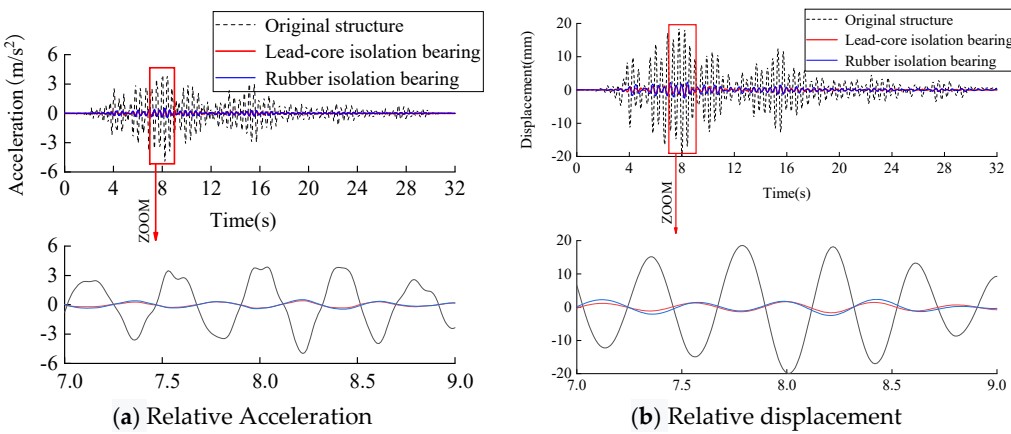

(**a**) Relative Acceleration        (**b**) Relative displacement

**Figure 12.** Damping effect of the isolated room.

Figure 13 shows the time history curves of the distance between the column of the seismically isolated room and the adjacent column for both types of bearing, with an initial distance of 200 mm, i.e., the width of the reserved seismic isolation joints. The results show that the minimum distance between the partial isolation room and the adjacent members was 167 mm for the lead-core rubber bearing and 108 mm for the natural rubber bearing, indicating that no collision occurred between the partial isolation room and the adjacent members under the action of either bearing, and that the partial isolation structure had a high safety reserve. Because of the greater stiffness of the lead-core rubber isolation bearing, the displacement of the partial isolation structure could be better controlled than with the rubber isolation bearing.

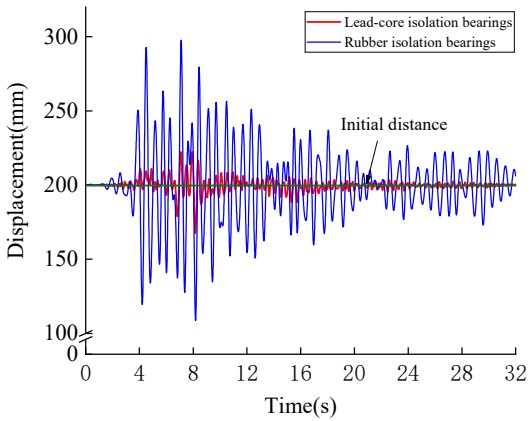

**Figure 13.** Distance between the isolated room column and adjacent column.

Figures 14 and 15 show that the overall seismic response of the structure was analyzed in terms of acceleration, displacement, inter-story displacement angle, and inter-story shear at the top of the structure under the two types of bearing, which had no obvious change with the original structure. Therefore, it was considered that the effect of changing the type of bearing on the overall seismic response of the original frame structure under this partial seismic isolation system was minimal and could be ignored.

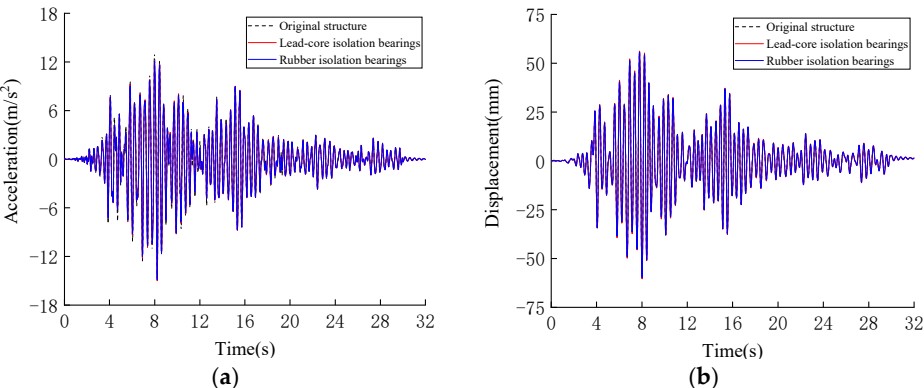

**Figure 14.** Time−history curve of the structure top (**a**) acceleration and (**b**) displacement.

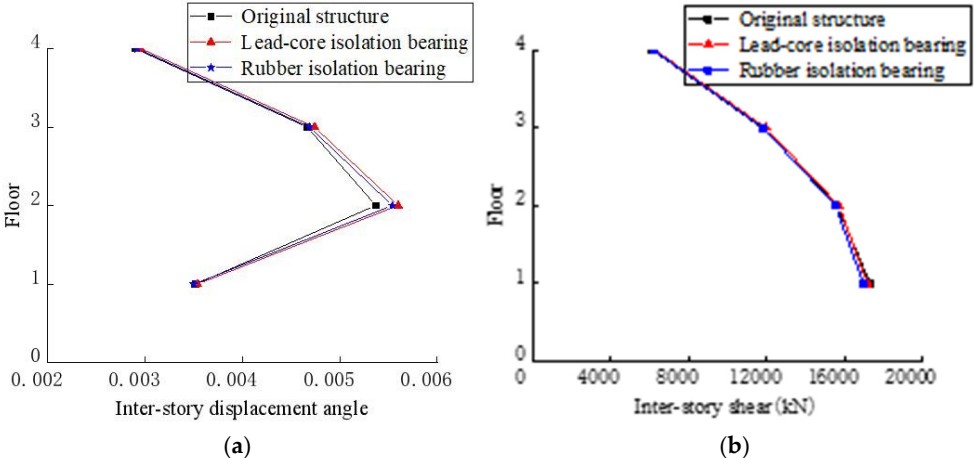

**Figure 15.** (**a**) Inter−story displacement angle and (**b**) inter-story shear.

### 4.2. Effect of Spatial Variations on Seismic Damping in Vibration Isolated Rooms

In order to apply the research results of the local isolation system to practical engineering, it was necessary to further explore the effect of the spatial variations of an isolation room on the seismic performance of the structure. On the basis of the original model, four additional isolated rooms were arranged on the second, third, and fourth floors, separately, as shown in Figure 16 below. The layout of the isolated room selected the corner, central, and two other characteristic locations on the floor. The analysis was helpful to find the rules of the optimal position of the isolated room in the structural design. The seismic wave inputs and bearing selection were consistent with the above, and a total of 12 partial isolation models were established.

Figure 17 shows the peak relative acceleration and displacement of the partial isolation rooms for each model. The analysis and calculation results showed that the relative acceleration damping rates of the isolated rooms located on the second, third, and fourth floors were 90%, 87%, and 83%, respectively, and the relative displacement damping rates were 89%, 85%, and 75%, respectively. It can be seen that as the location of the isolated room moved to the upper floors, the damping effect of the isolated room decreased, but was still much less than the seismic response of the corresponding room of the original

structure. Meanwhile, as shown in Figure 16, the in−plane variation in the location of the vibration isolated rooms had no significant effect on their damping effect.

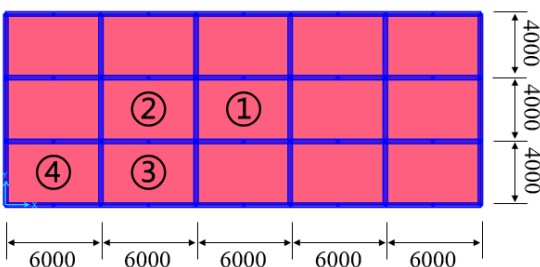

**Figure 16.** Floor plan (unit: mm).

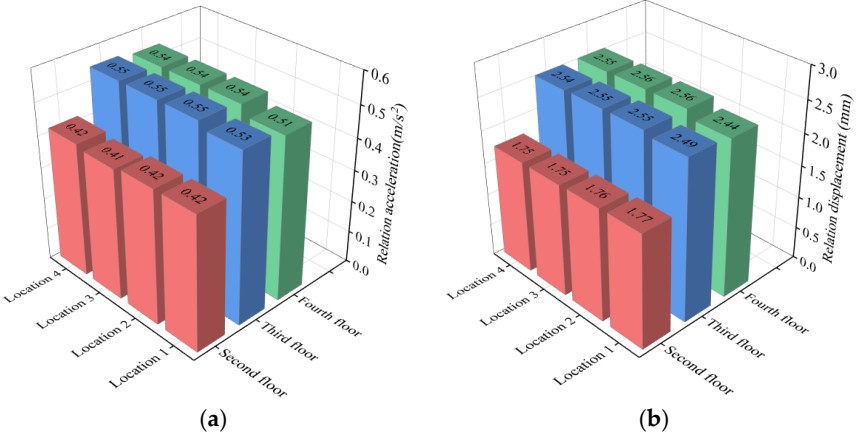

**Figure 17.** (**a**) Peak of relative acceleration and (**b**) peak of relative displacement.

Figures 18 and 19 show the peak acceleration and displacement at the top of the structure, the inter-story displacement angle, and the inter-story shear for the structure as a whole. It can be seen that the acceleration and displacement, inter-story displacement, and shear force decreased as the location of the isolated room moved to the upper floors. The best damping effect on the structure as a whole was achieved when the isolated room was located at the fourth floor, where the damping ratio of acceleration and displacement was about 10%, indicating that the elevated location of the isolated room effectively reduced the overall seismic response of the structure. The change in the in−plane location of the isolated rooms had no effect on upgrading the overall seismic performance of the structure.

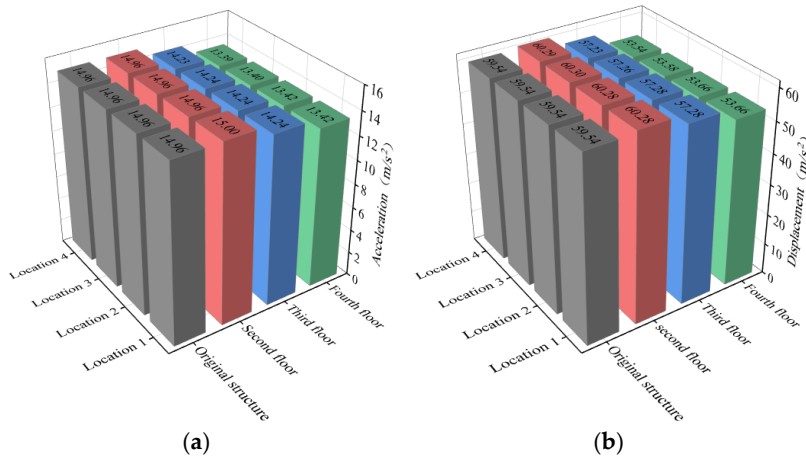

**Figure 18.** (**a**) Peak acceleration at the top of the structure and (**b**) peak displacement at the top of the structure.

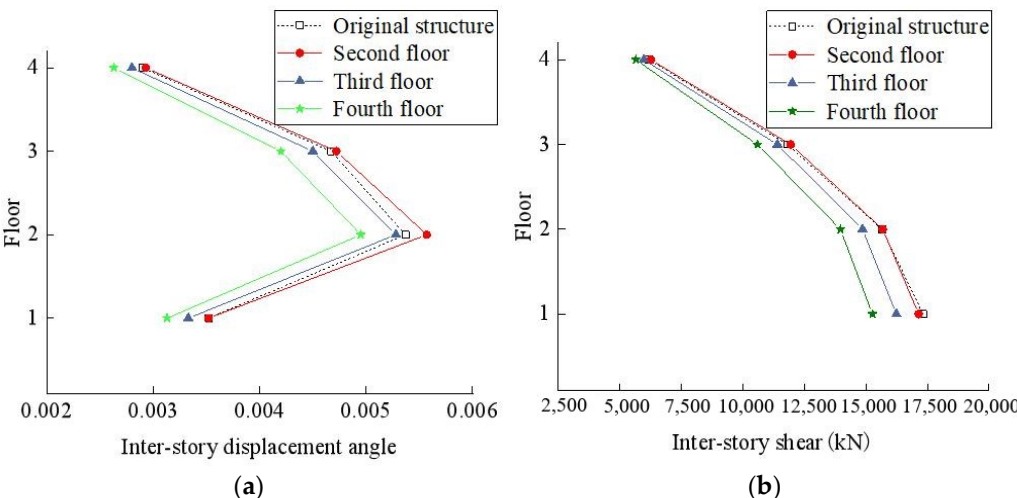

**Figure 19.** (**a**) Inter-story displacement angle and (**b**) inter-story shear.

### 4.3. Effect of Load Level on Vibration in Isolated Rooms

In addition, by increasing the floor load of the partial isolation room, it was expected that the damping effect would be similar to TMD. In addition, to further investigate the effect of partial isolation room loads on the seismic performance of the structure, the constant floor loads of the isolated rooms were adjusted to 3 kN/m$^2$, 6 kN/m$^2$, 12 kN/m$^2$, and 24 kN/m$^2$, respectively, and all of the conditions were the same as the isolated structure model in Section 2, except for the isolated room loads, and a total of four partial isolation models with different loads were established.

It can be seen in Figures 20–22 that as the partial isolation room loads increased, the relative acceleration and displacement of the partial isolation rooms increased, but were much smaller than the original structural seismic response. There was no significant change in peak acceleration, displacement, inter-story displacement angle, and inter-story shear at the top of the structure. Therefore, it was considered that changing the mass and load of the partial isolation rooms had little effect on the seismic damping effect of the partial rooms and the overall seismic response of the house, and could be ignored. At the same time, when the floor load of the partial isolation room was 24 kN/m$^2$, the deformation of the bearing and the shear and bending moment of the beam under the bearing were still within the safe range.

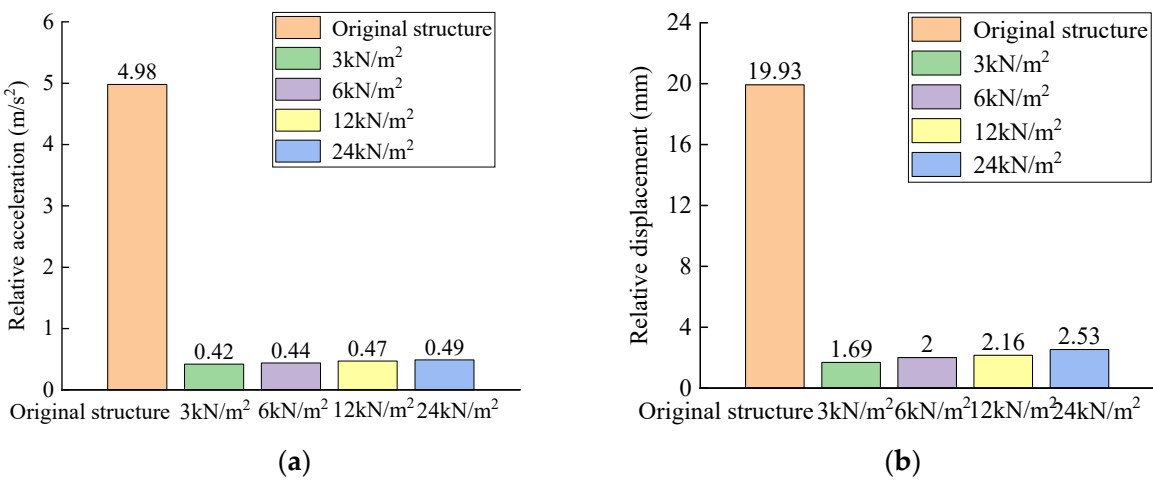

**Figure 20.** (**a**) Peak of relative acceleration and (**b**) peak of relative displacement.

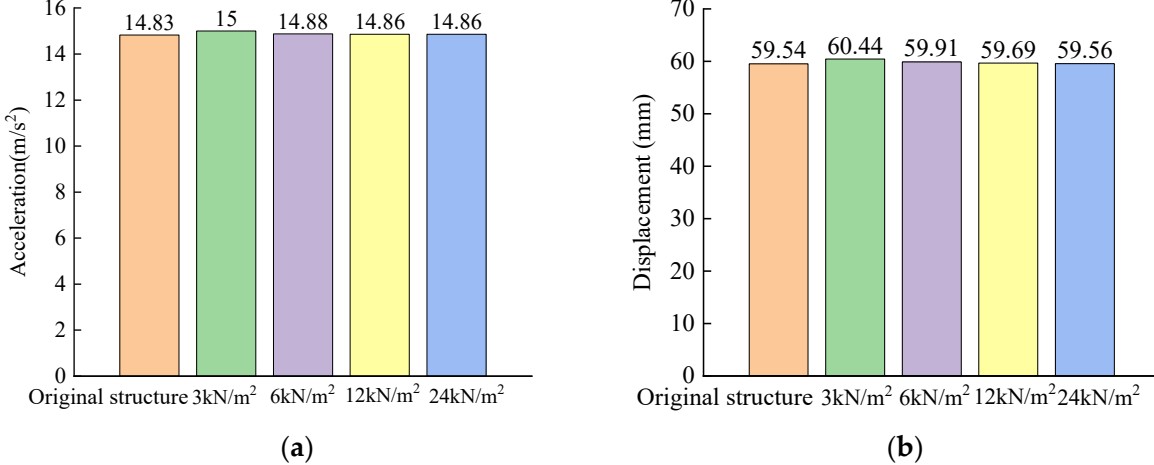

**Figure 21.** (**a**) Peak acceleration at the top of the structure and (**b**) peak displacement at the top of the structure.

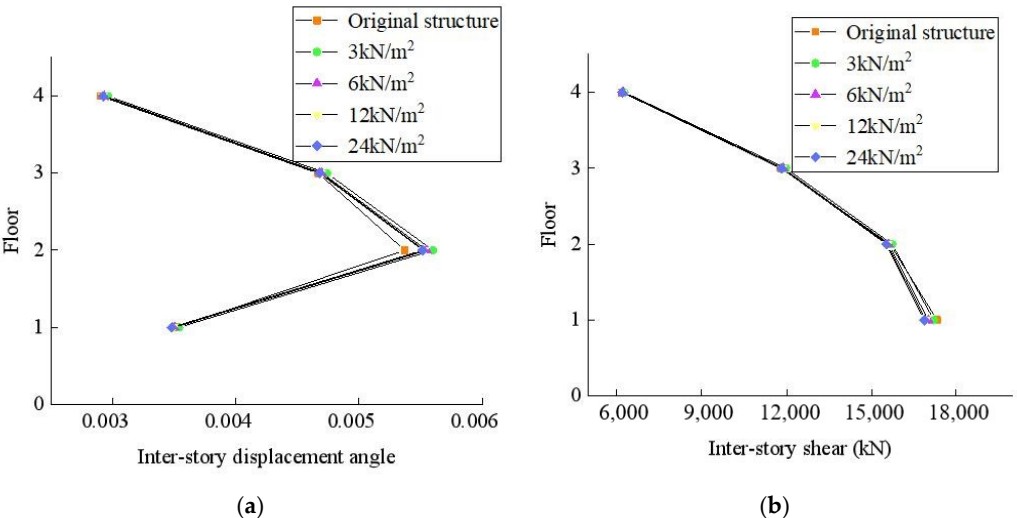

**Figure 22.** (**a**) Inter-story displacement angle and (**b**) inter-story shear.

## 5. Conclusions

This paper proposes a novel partial seismic isolation structure system and optimizes it by changing the type of seismic isolation bearing, the location of the partial isolation room, and the load. By comparing the seismic response of the original frame structure with the partial isolation structural model, the following conclusions are found:

(1) The partial seismic isolation system of the frame structure proposed in the study can significantly reduce the seismic response of the isolated rooms, with a relative acceleration and displacement reduction rate up to 90%, which is an obvious effect of seismic reduction. In addition, the partial seismic isolation system has a high safety reserve and has no effect on the seismic response and stability of the whole frame structure.

(2) The seismic performance of the structure was analyzed by changing the type of seismic isolation bearing and it was found that the lead-core rubber bearing could better control the relative displacement of the partial isolation room, and the deformation of the bearing was smaller than that of the natural rubber bearing, so the lead-core rubber seismic isolation bearing was chosen as more ideal.

(3) The relative acceleration and displacement of the seismic isolation room decreased significantly with the lowering of the floor position of the partial seismic isolation room, indicating that the developed partial seismic isolation system is more effective in reducing the seismic at lower floors. Moreover, the location of the partial isolation rooms can be

considered according to the functional objectives of the structure. In addition, changing the horizontal position of the partial isolation room has a limited effect on the seismic performance of the structure as a whole and on the partial isolation room.

(4) Changing the mass and load of the partial isolation room has a negligible effect on the damping effect of the isolated room and on the seismic response of the overall structure.

(5) The novel partial isolation system can significantly reduce the seismic response of the isolated room, but it has little effect on the overall seismic performance of the structure. More studies will be done to improve the seismic performance of both the isolated room and the structure. Moreover, the nonlinear behavior of the materials will be concerned under greater seismic loads.

In addition, there are some limitations in this study that need to be added in the follow-up work, as follows:

(1) Some assumptions proposed in the finite element numerical simulation may differ from the real engineering applications, among which the arrangement and design of the seismic isolation bearings are relatively simple, and the nonlinear response of the main body and the partial isolation structure is not sufficiently considered.

(2) In order to investigate the damping performance of the partial isolation design method under extreme load conditions, more seismic loads need to be added for effect verification, and the effect of near-fault ground shaking with impulsive components on the structure will be further analyzed in the subsequent study.

**Author Contributions:** Conceptualization, B.C.; Data curation, B.C.; Formal analysis, B.C.; Investigation, B.C. and J.X.; Methodology, B.C., J.X. and Y.X.; Project administration, J.X.; Resources, Y.L.; Software, Y.Q. and Y.L.; Supervision, B.C. and J.X.; Validation, Y.Q.; Visualization, Y.Q.; Writing—original draft, Y.Q.; Writing—review & editing, Y.L. All authors have read and agreed to the published version of the manuscript.

**Funding:** This work was financially supported by the China Earthquake Administration Basic Research Project (grant number 2018D18), the National Natural Science Foundation of China (grant numbers 51868048), and the Jiangxi Department of Education of Youth Science Fund (60221).

**Conflicts of Interest:** We confirm that the manuscript has been read and approved by all of the named authors and that there are no other persons who satisfied the criteria for authorship but are not listed. We further confirm that the order of authors listed in the manuscript has been approved by all of us.

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
