# Peer review of "Seismic Performance and Optimization of a Novel Partial Seismic Isolation System for Frame Structures"

_buildings, doi:10.3390/buildings12070876_

Round 1

Reviewer 1 Report

Comments and suggestions for authors in the attached pdf file.

Author Response

Thanks for your comments!Revised manuscript and responses are submitted in the attachments,red color is used in the revised manuscript to highlight the revisions.Please see the attachment.

Reviewer 2 Report

This paper presents an alternative partial seismic isolation system that can be used in concrete structures. In summary, the Authors demonstrated the advantages of the isolation system by implementing several response history analyses of a concrete building. In general, the paper is good. However, the Authors must perform the following revisions before proceeding further.

1.      By the end of the Abstract Section, please try to improve the discussion about the main findings and possible contribution of this paper.

2.      Right after Keywords, there is a section titled “0. How to Use This Template”. Such a section must be removed from the manuscript.

3.      In the first two paragraphs of the Introduction Section, the Authors are presenting a discussion about the main topic of the paper by documenting about 20 technical papers. This is good; however, it seems only as a general description of what was reported in those publications. I would recommend increasing the discussion about those papers. For example, advantages and disadvantages of the methods presented in those 20 papers must be documented, etc. Please revise this.

4.      By the end of the Introduction Section, I would recommend making emphasis on the main contribution of this paper to the Buildings Journal. In other words, the Authors must answer the following question: why does this paper deserve to be considered for publication?

5.      Please center Figure 2.

6.      Please include the plan and elevation view, respectively, of the building illustrated in Fig. 3.

7.      Please explain more in detail the hysteresis curve illustrated in Figure 4. How is such a curve implemented in SAP2000? Is such a curve validated through experimental testing of LRBs? Please justify this.

8.      In line 182 of the manuscript, please change “Chinses code” to “Chinese code”.

9.      As far as I understand, Section 2.4 of the paper is documenting the selection of characteristic ground motions. In this sense, two questions must be answered by the Authors: (1) are the ground motions presented in Figs. 5 and 6 accurate enough to be considered characteristic of the zone? Based on what is illustrated in Fig. 6, for long periods, they are not matching the standard response spectrum, why?, (2) why only three ground motions are considered? Most of the seismic-resistant codes recommend at least 7 or 11 ground motions to perform response history analysis, please justify this.

10.   Based on what is illustrated in Fig. 11(b), it seems that there is no big difference between the inter-story shear of the original and isolated structure. why is this happening? In certain way, the Authors are justifying this by the end of the first paragraph of Section 3.5, however, a more detailed explanation must be provided.

11.   Based on the results presented in this paper, it seems that the partial seismic isolation structure system is improving the seismic performance of the structure. However, it would be interesting to know a little bit more about the economic fact of implementing such isolation systems, are they very expensive? Have the Authors explored an economical study about its implementation?

12.   In the Conclusions of the paper, limitations of the study presented in this manuscript must be documented.

13.   In the References Section, please include DOI of every technical publication.

Author Response

(The authors gave the same response as above.)

Round 2

Reviewer 1 Report

The authors made the corrections. The paper can be accepted.